# Evaluation of Retention, Wear, and Maintenance of Attachment Systems for Single- or Two-Implant-Retained Mandibular Overdentures: A Systematic Review

**DOI:** 10.3390/ma15051933

**Published:** 2022-03-04

**Authors:** Raphaël Wakam, Aurélie Benoit, Kwamivi Bernardin Mawussi, Caroline Gorin

**Affiliations:** 1Unité de Recherche Biomatériaux Innovants et Interfaces, Université Paris Cité, 92120 Montrouge, France; pol.raphael.wakam@gmail.com (R.W.); aurelie.benoit@u-paris.fr (A.B.); 2Service d’Odontologie, Département de Prothèses, AP-HP/GH Paris Nord, 75018 Paris, France; 3Unité de Recherche Biomatériaux Innovants et Interfaces, Université Sorbonne Paris Nord, 92120 Montrouge, France; mawussi@univ-paris13.fr; 4URP2496, Pathologie, Imagerie et Biothérapies Orofaciales, Université Paris Cité, 92120 Montrouge, France; 5Service d’Odontologie, Département de Prothèses, AP-HP/GH Pitié Salpétrière, 75013 Paris, France

**Keywords:** ball and cylindrical attachment systems, implant-retained overdenture, retention force, wear, maintenance

## Abstract

Attachment systems (AS) enhance retention and stability by anchoring the overdentures to implants. Since 2002, the McGill consensus statement recommends the 2-implant-retained overdentures as the standard choice for edentulous mandible (2-IRMO). Considering the large number of AS available, it remains difficult for a practitioner to make a reasoned choice. A systematic review was conducted in PubMed/Medline and carried out independently by three authors, on retention, wear, and maintenance of AS used clinically or in vitro specifically for 1- or 2-IRMO. The 45 selected studies include 14 clinical and 31 in vitro studies. The risk of bias was evaluated according to the revised Cochrane risk of bias tool for randomized trials (RoB 2). The initial retention force of the cylindrical system is higher than the ball system. The retention loss, related to the wear of the retention device, is responsible for the most common need of maintenance, requiring activation or replacement. Plastic retention devices wear out faster and more significantly than metal ones, implying a worse time behavior of cylindrical systems, but their maintenance rate is similar. Neither system appears categorically superior. Cylindrical systems provide higher initial retention than ball ones; this advantage reduces over time with wear without affecting their need for maintenance.

## 1. Introduction

Different attachment systems (AS) with varied prosthodontic designs (stud, bar, magnet, double crown) and materials (metal and polymer) are used as primary or secondary retention devices in removable mandibular overdenture, retained or stabilized on implants [1,2,3,4,5,6]. According to the McGill consensus [3,6,7,8,9], or York Consensus Statement [10], the two-implant-retained mandibular overdenture (2-IRMO) is the standard treatment for the edentulous mandible. A more cost-effective alternative consists of an overdenture stabilized by a single midline symphysis implant (1-IRMO) [3,4]. Thus, even with a limited number of implant abutments, these AS provide better retention and stability [2,6,10,11,12], leading to a residual ridge height preservation [13,14] and a significant increase in chewing comfort and patient satisfaction [2,10,15]. Although IRMO is more cost-effective than a conventional prosthesis, it requires significant clinical maintenance because of wear-related retention loss of its AS and clinical needs of maintenance [5,13].

All AS for IRMO are composed of one male part, the patrix—an abutment connected to the implant—and one female part, the matrix—composed of a housing included in the intaglio surface of removable denture containing a replaceable retention device (RD) (Figure 1). When patrix and matrix are connected, retention is provided by friction between these parts whose behavior depends on the design and the constitutive materials of these components. In IRMO, to compensate for prosthesis depression on soft tissues, the resilient junction has been standardized with a calibrated space between the matrix and patrix to reduce the stresses on the rigid implants. The AS can be classified into two categories based of their abutment, which is either ball or cylindrical.

### 1.1. Ball Attachment Systems

The ball anchor was first proposed on dental abutments and then gradually adapted on implant supports. The most commonly used ball attachment system (BAS), Dalbo^®^ Plus (Cendres et Métaux SA, Biel-Bienne, Switzerland) [2], shows a satisfactory stable retention force overtime. However, its volume might limit clinical indications [16], even if it is available in different sizes to adapt to the vertical and horizontal prosthetic space. The most common patrix consists of a titanium alloy 2.25 mm diameter ball [1]. In contrast, the matrix is a titanium alloy or stainless steel case with a metallic or plastic retention device (RD) [2,5,13,16].

BAS show different types of RD such as the gold alloy lamellae or strips (Dalbo^®^ Plus or Dal-ro^®^ (Biomet 3i, Palm Beach Garden, FL, USA) [5,8] that need to be activated using a specific screwdriver and replaced after wear or retention loss and other systems that are nonactivable but replaceable such as the stainless steel spring rings Tima^®^ (Unor/Kaladent AG, Zurich, Switzerland) and TG-O-Ring^®^ (Cendres et Métaux SA, Biel-Bienne, Switzerland) [5,8], the rubber rings OP-Anchor^®^ (Inoue Attachments Co., Tokyo, Japan), O-Ring^®^ (Biohorizons, Birmingham, AL, USA) and Steri-Oss^®^ (Steri-Oss/Nobel Biocare, Zurich, Switzerland) [7,8,9], or the colored plastic RD (Southern^®^ (Southern Implants, Irene, South Africa), Ecco^®^ (Unor/Kaladent AG, Zurich, Switzerland)), Pro-Snap^®^ (Metalor Dental/Cendres et Métaux SA, Biel-Bienne, Switzerland)) and Preci Clix^®^ (Ceka Preci-Line/Alphadent, Waregem, Belgium)) [5,8] (Figure 1). In this AS, retention is ensured by friction between the RD and the patrix, mainly at the ball’s equator [17].

The manufacturers’ retention force ranges between 2 and 15 N depending on the chosen BAS. Although the initial retention is essential to immediately obtain the expected outcome for the patient, its stability in time is even more needed to avoid time-consuming maintenance and patient discomfort. The RD can be changed to retrieve its initial retention values in case of excessive wear and retention loss. BAS can function when implants are parallel or not. Indeed, they have been designed to accommodate changes in implant angulation up to 12, 15, 20, or 30 degrees [10,16,18,19]. The matrix components in the denture must remain parallel to the vertical path of prosthetic insertion [18].

BAS are the most widely used AS because they are easy to handle clinically, are relatively economical, and have a lower technique sensitivity [1,5,6].

### 1.2. Cylindrical Attachment Systems

Cylindrical attachment systems (CAS) were developed more recently to allow new AS clinical indications, especially in reduced prosthetic spaces, thanks to their smaller size and improved retention [2,6,20]. Developed in 1971, the first CAS was the Zest^®^ (Zest Anchors, Escondido, CA, USA) which evolved in ZAAG^®^ (Zest Anchors, Escondido, CA, USA). Nowadays, the Locator^®^ (Zest Anchors, Escondido, CA, USA) has become the most popular CAS with the lowest profile height, an improved retention force and, a design combining the best features of BAS, ZAAG^®^, and ERA^®^ (Sterngold Dental, LLC, Attleboro, MA, USA) [20,21,22,23,24]. When implant connections are compatible, CAS are preferred in situations of low prosthetic heights with a 2.5 mm AS [20,25,26,27]. However, they have a larger cross-section to maintain satisfying strength with a diameter of 4.1 mm [26].

For most CAS, the manufacturer considers (i) the cylindrical abutment as the « matrix » because of the groove in its center that allows a perfect fit of the RD acting as a push button, and (ii) the RD with its case as the patrix. For a better understanding and comparison of all AS, in this review, the names were standardized by calling the abutment the patrix, and the RD with its case included in the prosthesis the matrix (Figure 1). The Locator^®^ patrix consists of a yellow wear-resistant nitride-coated titanium cylindrical implant abutment featuring internal and external undercuts. The matrix consists of a cylindrical stainless steel case embedded in the denture basal surface and a polyethylene RD, which will ensure a resilient connection between the denture and the implant [7,15,25], with a vertical tolerance of 1.2 mm and a possible angulation of 8 degrees in all directions [7]. As for the ERA system (White/extra-low, Orange/low, Blue/medium, Grey/heavy, Yellow/high, Red/very high), the Locator^®^ RD are identified by color codes according to the required retention force and the angulation between implants [2]. The manufacturer claims that the retention force in this system ranges between 6.67 and 22.2 N. With standard RD (Blue/extralight, Pink/light, White/high), a difference of 10 degrees can be tolerated between the insertion axis of the RD and the central axis of the corresponding abutment. For interimplant angulations between 20 and 40 degrees, the manufacturer recommends extended RD (Red/extralight, Orange/light, Green/high).

Similar to BAS, once Locator^®^ patrix and matrix are interlocked, retention is ensured by friction between different surfaces [2,11]: (i) between abutment’s external surface and RD’s internal peripheral surface and (ii), optionally for standard RD, between the groove on top of the abutment and the RD’s internal central core. When the abutments are identically aligned parallel to the insertion path, retention is achieved uniformly from all the undercuts. However, in the case of angulation between them, friction will occur preferentially on the side presenting larger undercuts [25].

Since the ERA^®^ and the Locator^®^, other CAS have emerged over the past ten years. All use the same design principle: a titanium alloy implant abutment—with a specific coating surface—and interchangeable color-coded color RD—included in a case embedded in the denture basal surface. New designs are proposed to improve initial retention and wear resistance. Among these new AS, the Locator R-Tx^®^ (Zest Anchors, Escondido, CA, USA) shows an abutment with a pink wear-resistant titanium-carbon nitride coated surface, new undercut features, and four color RD (Grey/extralight, Blue/light, Pink/medium, White/high) without internal core and has been recommended for an interimplant angulation up to 60 degrees. For the Novaloc^®^ (Valoc, Möhlin, Switzerland), RD are made in polyetheretherketone (PEEK) to improve wear resistance and are inserted in titanium or PEEK cases. The system has six-color RD (Red/2.94 N, White/7.35 N, Yellow/11.77 N, Green/16.18 N, Blue/20.60 N, Black/25.01 N). The CM Loc^®^ (Cendres & Métaux SA, Biel-Bienne, Switzerland) was introduced to compensate interimplant angles up to 40 degrees and have both PEEK and precious alloy cases with four replaceable PEEK RD (Green/5.88 N, Red/11.77 N, Green/17.65 N, Blue/23.54 N). One clinical trial has evaluated the performance of CAS based on PEEK RD compared to the nylon RD of the Locator^®^ system [28], and three in vitro studies showed promising results regarding the long-term retention of PEEK RD [27,29,30].

#### Objectives

Considering the wide choice of marketed AS for IRMO, it remains difficult for a dental practitioner to make an appropriate clinical selection based on objective and scientific criteria. Manufacturers only provide partial data on their AS and rarely explain their testing conditions. In the literature, comparing the different studies is difficult because of the variability in their protocols and presented results. There is still no scientific data to support using one attachment system over another for the edentulous mandible [1,9,19]. Most AS are marketed without scientific and independent evaluation. This review compares the most common BAS and CAS used for 1- or 2-IRMO by assessing different criteria—initial retention and prosthodontic maintenance related to wear and retention loss—to establish the advantages and drawbacks of each system to assist the practitioner in making reasoned clinical choices.

## 2. Materials and Methods

### 2.1. Protocol

This systematic review (SR) was conducted following the Preferred Reporting Items for Systematic Review and Meta-Analysis (PRISMA) guidelines. The protocol was registered in PROSPERO (International Prospective Register of Systematic Reviews), at the UK’s National Institute for Health Research (NHS), University of York, Centre for Reviews and Dissemination, under the number: CRD42021265595. The PICO model for clinical questions was applied to structure the research question “Do Ball and Cylindrical attachment systems behave differently over time?”

Participants/population: Patients with completely edentulous mandibular arch mainly opposed to edentulous maxillary arch (or partial removable denture, or fixed denture, or natural teeth).Intervention/exposure: 1- or 2-implant-retained mandibular overdenture using ball attachment systems (BAS) or cylindrical attachment systems (CAS) only.Comparator/control: Comparison between CAS and BAS and, secondarily, their subgroups.Outcomes: Initial retention, retention after clinical use, retention loss, wear, and maintenance.

### 2.2. Search Strategy for the Identification of the Studies

An electronic search on PubMed/Medline was carried out on 8 September 2021 using different keywords: “Denture Precision Attachment”[Mesh], “implant attachment*” OR “overdenture attachment*” OR “attachment system*” OR “stud attachment*” OR “resilient attachment*” OR “spherical attachment*” OR “ball attachment*” OR “cylindrical attachment*” OR “locator attachment*”, maintenance OR complication* OR prosthodontic* OR prosthetic* OR prosthes* OR retention OR wear, “implant stabilized overdenture*” OR “implant retained overdenture*” OR “implant supported overdenture*”. The search included language (English) and publication status restrictions (Abstract) with date limitation (1996–2021). In addition, a manual screening was carried out among the references of selected articles to gather further relevant papers. Three review authors (R.W., A.B., and C.G.) assessed studies independently for eligibility by initially screening successively titles and abstracts (Figure 2). Then, the full-text articles were retrieved for further assessment when studies met the inclusion criteria. Studies that had insufficient data were excluded. Any disagreements were resolved by discussion between the two authors.

### 2.3. Eligibility Criteria

Different inclusion criteria have been used:(i)in vitro studies reporting on initial retention, retention loss, and wear of real or simulated 1- or 2-IRMO using stud-shaped BAS or CAS,(ii)clinical studies (i.e., prospective, retrospective, randomized controlled trials including cross-over trials) reporting on maintenance of AS (retention loss, wear) in 1 or 2-IRMO regardless of follow-up time.

The exclusion criteria were:(i)maxillary overdenture,(ii)partial dentures,(iii)fixed overdentures,(iv)overdentures on remaining natural teeth or roots,(v)more than two implants,(vi)implant splinting,(vii)other AS (bars-clips, magnets, telescopic double crowns),(viii)anecdotal BAS and CAS.

Comparative studies were rejected if only one type of BAS or CAS was used versus other AS. Case reports, reviews, and short communications were rejected.

### 2.4. Outcome Measures

The primary outcome measures concerned:(i)the initial retention and its evolution over time of clinical use (Newton (N), loss of retention (%)), in correlation with the observed wear patterns (scanning electron microscopy with qualitative description) and the dimensional changes (μm) of AS components,(ii)the maintenance follow-up (frequencies: months/years), and maintenance procedure of AS (activation/replacement).

The secondary outcomes were to report:(i)the effects of denture cleansing solutions on the retention and wear of AS,(ii)the overdenture maintenance (i.e., reline/rebase of denture, fracture repair or occlusal adjustment, denture replacement),(iii)the interimplant distance (mm) or angulation (°) and their impact on primary outcomes.

### 2.5. Data Extraction and Analysis

For eligible studies, data were extracted and collected in an Excel spreadsheet by one review author (R.W.) and checked by two others (A.B., C.G.) independently. In case of disagreement, a consensus was obtained with discussion between the authors. Finally, 45 articles have been identified out of a total of 1134 selected among which 14 clinical and 31 in vitro studies.

### 2.6. Study Quality Assessment

The quality assessment of each clinical and in vitro study, based and adjusted from the revised Cochrane risk of bias tool for randomized trials (RoB 2) [31] was independently evaluated by two review authors (R.W., A.B. for in vitro studies, and R.W., C.G., for clinical ones). Differences were resolved after team discussion.

For the clinical studies, the risk of bias (RoB) was assessed by the following items: randomization process, deviations from intended interventions, missing outcome data, measurement of the outcome, selection of the reported result, and other biases such as the presence of a control group, description of inclusion criteria, the status of the maxillary arch (complete edentulous and overdenture), years of follow up (≤1, low; ≤2, middle; ≥3, high), follow-up planning. For other biases, the study received a ‘yes’ if at least 4 parameters were respected over 5, a ‘middle’ if only 3 parameters were respected and a ‘no’ otherwise. RoB 2 criteria were adapted for in vitro studies, assessed by the following items: randomization process, blinding of the test operator, post-processing of experimental data (sample size and statistical analysis), and detailed protocol for retention and wear measurements.

Each item was evaluated following RoB 2 recommendations. Finally, to assess the overall RoB, a study with at least one ‘no’ was classified as ‘high RoB’, a study with ‘unclear’ or ‘middle’ for one or more items was classified as ‘unclear RoB’ (except in both cases for other biases for clinical studies and blinding of the test operator for in vitro studies). A study with ‘yes’ in all domains was classified as ‘low RoB’.

Among the 45 selected studies, 25 studies showed a low RoB and 13 moderate RoB. Only 7 clinical studies exhibited a high RoB, primarily due to their lousy score for randomization. In vitro studies were poorly scored for the blinding of test operators, but it did not impact their overall RoB (Figure 3 and Figure 4).

## 3. Results and Discussion

According to the aim of this SR, the results are presented in four parts: (i) the initial retention of BAS and CAS, (ii) the retention loss and wear of AS correlated to the prosthodontic maintenance; the influence of implant parameters (iii) and experimental conditions (iv) on the retention, wear, and maintenance. Finally, paragraph (v) exposes the limitations of our study and, in particular, the difficulty of comparing studies directly because of the variability in their protocols and reported results.

### 3.1. Initial Retention of BAS and CAS

Retention is defined as the maximum force developed until dislodgement of the denture from its mucosal and/or implants bearing surfaces. It represents the mechanical resistance to displacement along the insertion axis opposite to denture insertion [27]. Retention must be high enough to limit unwanted movements of the removable prosthesis while inducing reasonable lateral forces on the implant, whatever its size or angulation [19]. The adequate and satisfactory retention force for each patient remains challenging to assess [4], the minimum commonly accepted lying between 5 and 20 N while being maintained over time [11,13,14,27]. Some authors suggested that 5–7 N are sufficient for 1-IRMO [13,14,32,33], and other investigators proposed 10 N [33] or 20 N for 2-IRMO [11]. In vitro studies showed that most AS in 1 or 2-IRMO could reach this acceptable range.

BAS show retention forces in agreement with manufacturers’ data, in the range of 2–16 N for 1-IRMO. However, regarding CAS for 1-IRMO, a wide range of retention forces is available through the different color-RD between 3.84 N and 16.6 N for Locator^®^ and 13.12 and 24.03 N for ERA^®^ [21,23] (Table 1).

For 2-IRMO, initial retention is generally equal to or greater than twice the retention provided by a single AS. It ranges between 10.6 N and 56.2 N for BAS and between 9.95 N and 108.9 N for CAS (Table 1) depending on the chosen AS and RD (9.95–108.9 N for Locator^®^, 20–75 N for Locator R-Tx^®^, 57 N for Novaloc^®^, 12.45–44.07 for CM Loc^®^, and 12.7–35.24 N for ERA^®^). Higher retention may be related to unwanted misalignment between implant axes or between implants and insertion axes due to inaccuracies in implant positioning and in controlling the direction of dislodgment in vitro. Despite many studies (ERA^®^ vs. Locator^®^ [34], Locator^®^ vs. CM-Loc^®^ [27], Novaloc^®^ vs. Locator^®^ R-Tx vs. CM Loc^®^ [30], Locator^®^ vs. Locator R-Tx^®^ [35]), a direct comparison of initial retention between different CAS remains difficult because of their large choice of RD.

Several in vitro studies have compared the initial retention force of BAS and CAS. Five studies [5,13,21,32,36] presented in Table 1 observe greater retention forces for CAS (especially Locator^®^ and ERA^®^) over BAS (OP Anchor^®^, O-Ring^®^, Pro-Snap^®^, Ecco^®^, and Dalbo Plus^®^) even if the values vary between studies. This could be explained by larger friction surfaces observed in CAS between the cylindrical abutments and the RD compared to the linear contact located at the ball’s equator in BAS. Three contradictory studies found greater retention for BAS with plastic RD [11,33] or metallic lamellae [16] compared to the Locator^®^. These results could be explained by the use of a larger implant (8 mm) with AS diameter (5.9 or 7.9 mm) [11], anecdotic in our clinical context, and a considerable variation in standard deviations which raises questions on the methodology applied (40.3 ± 15.83 N for Dalbo Plus^®^ and 33.5 ± 9.77 N for Locator^®^) [16].

In conclusion, the few comparative in vitro studies agree that the initial retention force of CAS (Locator^®^) is more significant than BAS, even if the tested RD and the experimental conditions vary between studies. However, this superiority does not seem to be felt clinically, according to Krenmair et al. that showed similar patient satisfaction for the two types of attachment during 3 months of wearing [6].

### 3.2. Maintenance, Wear and Loss of Retention of BAS and CAS

The maintenance of an IRMO encompasses a multitude of parameters that are not all specific to the presence of AS in prosthetic construction and vary considerably between studies (Table 2). It concerns male and female parts of AS, mandibular overdenture, opposite prosthesis, peri-implant or soft tissue-related complications, and quality of life. To provide a good integration of the prosthesis over time, dental practitioners need to regularly modify its shape in order to adjust over-extensions and dental occlusion [12] or repair the prosthesis (overdenture replacement, fractures, removable relining procedures) [3]. Maintenance of the AS generally occurs in the first year after the prosthetic insertion [2,8,17] and is more prevalent during the first 4 years [8]. The most common complication is the loss of retention, mainly due to wear of the RD generated by repetitive mechanical loading, which requires its activation [8,37] or replacement after a mean time estimated at 11.2 months [3,7,38]. This event affects all patients, and it can occur up to 7 times per AS after 10-year follow-up in 1-IRMO [39]. RD of BAS can be activated (gold alloy) or replaced (rubber ring, nylon or plastic cap, circular stainless steel spring), whereas RD of CAS can only be replaced (polyethylene or PEEK caps). Other complications could be the management of abutment or RD loss or the fracture of the male and female parts, which remain exceptional [3,8,16,19]. According to some authors [8,9,12], AS is considered successful if there are no more than two activations, repairs, or replacements of either patrix or matrix in the first year of clinical use. Considering the overdenture, success is characterized by no more than one relining to improve fit and stability.

Different paradigms correlate in vitro mechanical loading with an estimated wearing time. Some studies use insertion–removal cycles (IRC) with different numbers of cycles per day: 3 [12,40], 4 [16,41,42] or 5 [21]. Others consider only chewing cycles (CC) [13] or a combination of CC and IRC [27]. Depending on the study, one year of clinical use would be equivalent to 1000–1800 IRC or 400,000 CC. Based on this estimation, some in vitro studies [5,29] characterized retention loss and wear of the AS for a number of cycles equivalent to more than 15 years of service, that far exceeds the mean time between two activations or replacements estimated at 11.2 months [3,7,38]. An equivalent wearing time is systematically indicated in the following presentation of in vitro results to highlight this correlation.

Wear is defined as material loss due to contacts between the matrix and the patrix, which increases the gaps between the different parts [2,41]. In the oral cavity, wear, mainly caused by mechanical loading (chewing and insertion/removal of the prosthesis), is located more in the labiolingual direction than in the mesiodistal one, and is increased when implants are inclined with respect to the insertion axis of the prosthesis [17,29,41]. Besides mechanical factors, environmental factors, including temperature change or aggressive bathing solutions, can also alter AS, especially polymeric RD [29]. Sometimes, damages are caused by the tools used to insert, remove, or activate the RD in the matrix housing during maintenance [13]. In the short or medium term, studies have shown that wear occurs at the equator of titanium abutments discreetly and similarly with no measurable change in diameter regardless of the AS [3,5]. Different surface wear patterns (Table 3) appear according to the type of RD and its constitutive material [4]. For all AS, RD was designed in the less rigid material to wear out preferentially, its replacement being more manageable and less expensive, leaving the abutment (patrix) intact. All analyzed clinical and in vitro studies conclude that loss of retention over time concerns both BAS and CAS but in different proportions [6,7,13,36,41]. The IRC and/or CC, performed by the patients in clinical studies or automatically in in vitro studies (fatigue tests) gradually decrease retention (Table 1).

Concerning BAS, retention is ensured mainly by the friction between two metallic surfaces (titanium patrix/precious alloy or stainless steel matrix) and sometimes between metallic and polymeric surfaces (titanium patrix/polyacetal or rubber ring matrix). Wear including abrasion and scratches can exceptionally affect the patrix—diameter reduction up to 25% observed with an anecdotic titanium RD after 5000 IRC (3 years) [43]—but affects the matrix essentially [6,7,8,9,15,17,41,44].

Clinical studies about BAS showed that wear of the precious alloy RD in contact with the titanium alloy ball abutment generated blunt lamella edges with a loss of RD thickness (1 year: 7 µm, 3 years: 47 µm, and 8 years: 70 µm) [41,44], and gold or titanium deposits on the RD [15,41]. Two clinical studies even described a significant wear of a titanium ball abutment (patrix) with precious alloy RD, increasing between the first and third years and remaining stable until 8 years [4,17], characterized by a flattening of the surface [15], an eccentricity of the ball in the long term and a reduction in diameter at the equator (1 year: 5–7 µm, 3 years: 19–22 µm, 8 years: 22–31 µm) [41]. Scratches were also observed in vitro [5], after 50,000 IRC (30 years) on ball abutments associated with metallic RD. It appears that the combination of titanium alloy ball with precious alloy RD or with plastic cap remains the most favorable configuration overtime to limit the loss of retention or to reduce post-insertion aftercare. Indeed, their initial and final retention are similar, although a retention increase (up to 65% of the initial retention) is observed up to 15,000 CC (2 weeks) or 500 IRC (4 months), which is maintained in the case of IRC until 5500 IRC (3 years) [45] or followed by a slight decrease in case of CC to return to its initial value at 100,000 CC (3 months) [5,13,16,21], maintained until 400,000 CC (1 year) [5,8,13].

Concerning CAS, friction involves only metal and polymer surfaces (titanium patrix/nylon or PEEK matrix). In vitro [21,25] and clinical [4] studies showed excessive wear of the nylon RD of Locator^®^ AS with the presence of mineral deposits coming from the titanium alloy abutment [15], that required RD replacement, the abutment presenting little or no abrasion [3,5,11,13,21,27]. The dimensional changes and deformations prevailed on the central core of Locator^®^ RD (loss of substance and surface irregularity) [21,25] and ERA^®^ nylon RD (>5.0% surface smoothing) [21], compared to their periphery (2.1%) or the metallic matrix housing (0.9%) [21]. The wear pattern of PEEK RD (Novaloc^®^ or CM Loc^®^) is not well established: one in vitro study showed severe deformation after 30,000 IRC (16 years) [29], while another none after 400,000 CC (1 year) [27]. ERA^®^, Locator^®^ and CM Loc^®^ RD decrease significantly in long term retention, and in particular for Locator^®^ (66 to 75% of their initial retention) compared to CM Loc^®^ (32 to 47%) after 30,000 IRC (16 years) [29], and this appears earlier for ERA^®^ RD (85% after 1000 IRC/1 year) [21,23]. In general, for Locator^®^, a significant decrease in retention is even observed during the first 10 to 20 IRC, but retention loss remains not significant after 100,000 CC (3 months) [13,46]. Nevertheless, a progressive decrease is observed until 1000 to 2000 IRC (i.e.,1 to 2 years of clinical service) [5,10,13,35,41]. Concerning other CAS, no significant difference between initial and final retention was found at 5000 IRC (3 years) for CM Loc^®^ [27,29] and at 10,000 IRC (6 years) for Novaloc^®^ and Locator R-Tx^®^ [30].

The remarkable mechanical properties of PEEK (tensile and bending strength, fatigue behavior) [27] could explain the increased performance of Novaloc^®^. Also, its RD design present a vertical slit that expands during the IRC or CC, thus reducing the deterioration of the material. In addition, for Novaloc^®^ and Locator R-Tx^®^, manufacturers provide color-RD associated with different levels of retention and wear behaviors. The variable selection of RD among studies can explain differences observed in wear and retentive properties.

Comparative clinical [4,6,7,15] and in vitro [9,13,32] studies between BAS and CAS concluded that ball precious alloy strips RD presented lower surface wear, and needed less or equivalent post-insertion maintenance than Locator^®^ RD. Only Cristache et al. [9] found a higher maintenance requirement for BAS after 5 years, using an obsolete version of gold alloy RD. Even if the experimental conditions are different (study design, number of implants, number of cycles, follow-up period), all except two [9,32] agree that Locator^®^ loses more retention than BAS [3,6,15,21]. This decrease evaluated up to 40% after 400,000 CC (1 year) [13], and to 66–75% after 30,000 IRC (16 years) [29] is mainly attributed to the nylon RD wear, which is more frequent in this system both in vitro and clinically [7,13]. In one clinical study, retention loss is evaluated at 70% of initial retention for the Locator^®^ (including 50% in the first 3 months) and only 40% for the ball after 2 years [15]. Conversely, the retention of BAS is more stable over time, with a nonsignificant decrease after 400,000 CC (1 year) (less than 10% of the initial retention force) [13]. Nevertheless, despite a more significant loss of retention for Locator^®^, its retention force, after 1 to 2 years of wearing, remains higher [13] or similar [5] to BAS.

In conclusion, contact surfaces must be made in different materials (polymer RD/titanium abutment or precious alloy RD/titanium abutment) to limit wear and retention loss. Abi Nader et al. [13] estimated that the Locator^®^, which has higher initial retention, would lose its superiority over the ball after 300,000 CC (9 months), leading to a nonsignificant difference in the final retention between the two AS, after one year of clinical service (400,000 CC). This difference might have a clinical relevance according to the initial retention needed and the interval between two maintenance sessions (activation or replacement of RD) [16]. Both clinical acts require minimal chair-time [6]. The practitioners will then choose the AS depending on the frequency of maintenance appointments, the initial retention force needed, and their usage preferences.

### 3.3. Influence of Implant Parameters on Retention, Wear and Maintenance of AS

The previously analyzed in vitro studies only concerned parallel implants placed under optimal conditions. However, initial retention, loss of retention, wear, and postinsertion aftercare would occur much sooner in a clinical context. The prosthetic construction on the opposing arch (removable or fixed overdentures, full or partially dental arch), prosthetic hygiene, regular dental aftercare sessions and implant parameters (implant angulation, interimplant distance, angulation between implant and AS) can easily be tuned in in vitro studies. Their impact on retention and wear of AS is presented in the following sections.

According to the clinical context, when a 2-IRMO is indicated, the two implants should be placed (i) parallel to the vertical path of insertion and (ii) parallel to each other in the interforaminal region (Figure 5). However, due to anatomical constraints or surgeon experience [10,25,47], the ideal orientation of implants is sometimes impossible to reach, and angulation is often observed between the two implants [47]. Indeed, without a surgical guide, experienced practitioners manage to place the implants with an interimplant angulation of 4.6 ± 2.9 degrees in the coronal plane and 3.5 ± 2.6 degrees in the sagittal plane [15]. Moreover, an angulation between implants of more than 6 degrees in the sagittal or 6.5 degrees in the frontal plane requires a significantly higher number of denture adjustments [10] and affects AS retention and wear. Although it remains clinically tricky to achieve a 0-degree angle, practitioners should try to reach for the lowest interimplant angulation [25], especially projected in the sagittal plane because the wear is more important in the labio-lingual direction [15]. It is difficult to categorically state the influence of implant or interimplant angulation (α and β, Figure 5) on retention for any type of attachment, notably for CAS because of the scattering of the results presented in Table 4. This discrepancy may be related to the use of four chains connected to the prosthesis to impose its vertical displacement in most studies [27,33,47,48]. The orientation of the insertion/disinsertion axis may vary between each test, affecting the measured retention force and modifying the intended implant angulation. Perfect control of the dislodging axis can only be ensured by a rigid connection between the prosthesis and the loading machine or by actual measurement of the chains’ orientations for each cycle.

For interimplant angulation between 20 to 40 degrees, the Locator^®^ manufacturer recommends using extended RD, which provide similar retentive behavior as standard RD used between 0 and 20 degrees. However, it is important to realize that using standard RD for huge interimplant angulations (60°) generate excessive initial retention force but rapid RD wear, due to the larger undercuts created by implant angulation [10,25,27,35,47,48]. No effect of interimplant angulation was observed up to 40 degrees on CM-Loc^®^, Locator^®^, Locator R-Tx^®^ and Novaloc^®^ [30,35].

Concerning BAS, until 30 degrees, implant angulation seems to have little impact on their initial retention and loss of retention [10,19,27,49]. However, some authors reported a reduction in their retention force up to 25% increasing implant angulation from 0 to 30 degrees. Regardless of the implant angulation, their initial retention is significantly lower than CAS, and their retention loss remains negligible (10 to 18% for BAS vs 80% for CAS) [10,19,27]. When implants are not sufficiently parallel, one study [18] highlights the importance of parallelizing the RD and aligning them with the prosthesis insertion path to guarantee sufficient retention. Indeed, parallel RD on parallel implants and parallel RD on nonparallel implants have no significant difference in retention (20.11 N at 0 degrees, 21.21 N at 10 degrees, or 18.78 N at 15 degrees).

No studies were found on CAS regarding RD angulation, but clinical recommendations consist of aligning them with the insertion path.

In IRMO, as implants are positioned in the interforaminal or intercanine area, the consensus establishes that the interimplant distance must range between 15 and 30 mm (Figure 5). According to anatomic limitations and implant space requirements, the mean distance is 22.88 mm [14], and no significant impact was found on the prosthetic construction using different distances (15 mm [13], 20 mm [18,22], or 22 mm [10,32,47,48]). Two in vitro studies [14,33] observed retention sufficient for clinical use for CAS and BAS for different interimplant distances covering the recommended range. Therefore, Practitioners can determine implant position according to the clinical situation without affecting prosthesis retention.

### 3.4. Influence of Experimental Conditions on Retention, Wear and Maintenance of AS

Several experimental conditions can affect retention and wear observed in in vitro studies. Indeed, the use of a real denture or simulated denture blocks, the dry or wet environmental conditions, the dislodgement speed, the loading conditions, and the characteristics of the testing machine (precision, sensitivity, margin of error) could impact the obtained results.

Although mechanical stress is considered to be the main factor in wear and retention loss of AS, their varied and humid environments (saliva, cleansing products, mouthwashes) have been studied and show little impact on AS retention and wear (Table 5) [22,26,40,50,51]. The main components of several brands of denture cleaners are sodium bicarbonate, sodium perborate, potassium monosulfate, citric acid and ETDA, and only a discoloration was observed on polymer RD with NaOCl solution [26]. Moreover, NaOCl and sodium bicarbonate significantly affect the retention of Locator^®^ after 6-month [26,40,50] and after 12-month immersion time [26,50,51]. The only conclusion that can be drawn is not to immerse AS, especially those composed of polymer RD, in highly reactive chemical solutions (NaOCl) for a long time and to rinse them well after use. Thus, the practitioner should advise patients on the daily maintenance of their prostheses. Prosthesis immersion should be limited to a few hours per week in an appropriate cleansing solution; a regular cleaning using a toothbrush and mild soap after each meal is sufficient and cost-effective.

All studies summarized in Table 1 and Table 3 show variable experimental conditions to simulate in vitro wear of a prosthesis and thus to measure the wear and retention loss expected clinically over time. Most studies [10,16,21,40,41,42] used IRC, one study [13] considered CC, and another one [27] a combination of both cycles. Indeed, the observed wear depends on the chosen fatigue cycle applied to any AS. These cycles applied in vitro are idealized and cannot perfectly represent the complex mechanical loading of an AS in the oral cavity, such as overdenture rotations, excentric chewing, or tilting of the prosthesis before disinsertion. However, all studies show a correlation in the increase in retention loss, wear, and maintenance with the number of cycles and consequently with the time of clinical use. In addition, the cyclic dislodgement rate (cycles/min) could affect the measurement of retention and wear, especially for polymer RDs that exhibit viscoelastic, strain-rate sensitive mechanical behavior. After a dislodging cycle, these materials require time to return to their original shape. The stress magnitude induced in the RD during dislodgment—and thus the retention force—also depends on the disinsertion speed. Most in vitro studies have used a dislodgement rate of 10 cycles/min [21,27,42,47,48] but it can vary between 6 [11] and 40 [14] cycles/min. It has been reported that, the retention force of the Locator^®^ does not depend on the rate up to 20 cycles/min but decreases significantly at higher speeds (40 cycles/min) [14].

Some authors have observed variations in the retention force within a range of identical attachments [42], as well as differences in size or composition between different batches of the same product [18]. These variations could be related to discrepancies during the manufacturing process and are therefore not controllable or predictable by dental practitioners.

### 3.5. Limitations

This SR aims to compare the most common BAS and CAS used for 1 or 2-IRMO to assist the practitioner in making reasoned clinical choices. We consider that initial retention and prosthodontic maintenance related to wear and retention loss are critical parameters for the final decision. Thus, the PICO question comprises 5 different outcomes (Initial retention, retention after clinical use, retention loss, wear, maintenance), all related but rarely evaluated in a single clinical or in vitro study. Both types of studies were included in this SR to explain the need for maintenance observed clinically by wear and retention loss mostly observed in vitro. Selected studies present substantial variability in their designs, with numerous parameters concerned:(i)prosthetic parameters (real or artificial denture, prosthetic design, implant number, interimplant distance, and angulation between implants or between implants and AS),(ii)in vitro experimental parameters (environment, loading conditions, and sample characteristics),(iii)clinical conditions in patient studies (bone loss, type of prosthetic construction on the opposing arch, dental occlusion, masticatory forces, prosthetic hygiene, and regular dental aftercare visits after setting denture).

The heterogeneity of reported data also induces missing information in summary tables and limits the quality of evidence of this SR, direct comparison between results being difficult. No meta-analysis could be performed given the low confidence expected due to the lack of standardization of the selected studies.

Additionally, the quality of clinical studies has been assessed as low using RoB 2 criteria, with 7/14 studies at high and 3/14 at moderate risk of bias. In particular, they present a high (4) or moderate (9) risk for other biases summarized in Section 2.6, explicitly introduced for their significance in our SR question (including follow-up, maxillary status, control group and inclusion criteria). The use of the Grading of Recommendations Assessment, Development, and Evaluation (GRADE) approach to assess the certainty of evidence from clinical trials could not be performed secondary to the inability to conduct a meta-analysis. To compensate for their low quality of evidence, clinical results were systematically compared with in vitro results when possible. The quality of in vitro studies, assessed with adapted Rob 2 criteria, appears acceptable, with 22/31 studies at low and 9/31 at unclear risk of bias. However, it should be noted that data heterogeneity was substantial, mainly due to protocol variability, especially regarding the control of implant position and insertion/disinsertion axis orientation.

Finally, the studies selected could not fully address the SR question, but they highlighted trends supported by both in vitro and clinical studies. Every identified contradictory result was explained by further analysis of the study design and its bias. The following conclusion summarizes the significant findings, presents guidelines to improve in vitro studies on AS, and clinical recommendations for practitioners.

## 4. Conclusions

Given the difference in the interlocking mechanisms, the variability of the studies procedures and the lack of standardized technical protocols, a direct comparison of the different AS remains impossible to determine the best AS in 1 or 2-IRMO. BAS remain the most evaluated AS in the literature due to their anteriority, whereas Locator^®^ is now the most used. According to the objectives of this SR, three conclusions can emerge from the studies analyzed.

First, the initial retention forces of CAS are higher than those of BAS, but their resistance to fatigue (retention stability) is much lower, related to the nylon RD wear. After a certain amount of IRC and/or CC, the retention force provided by the Locator^®^ could become lower than the retention force of BAS, but the time threshold remains difficult to determine.

Secondly, implant abutments in each system are barely affected by wear, unlike the RD that show significant plastic deformation, especially in the central core of CAS (Locator^®^). In contrast, the metallic lamellae RD of BAS present relatively moderate wear at the base and discreet deformation at the top.

Thirdly, the retention loss is correlated to the wear of each RD and justifies regular maintenance, especially during the first year, leading to RD activation (BAS) or replacement (CAS), according to the AS. No significant difference was found in the overall maintenance incidence rate of each RD. Each practitioner will thus choose a system according to its patient needs and clinical situation. BAS initial retention is less effective but more durable than CAS, whereas CAS are more retentive but require a regular change of their RD to maintain their retention superiority.

For in vitro studies, the following guidelines can be proposed to improve their repeatability and enable comparison between them. To characterize the retention of an AS in standard conditions, repeated measures on a single AS should be preferred. The tested AS must be identified: abutment, chosen retentive device, matrix housing. Blocks containing male and female parts should be carefully manufactured to ensure parallelism (or the required angulation) between the AS axes and the testing machine. Force measurements should be performed with an appropriate load cell (100–500 N), with a dislodging speed (50 mm/min), and in a solution (artificial saliva) representative of the clinical use. Preliminary tests should be performed to set the pre-load value necessary to ensure complete interlocking of male and female parts before dislodgment, especially for polymeric RD. A relaxation time can be added for polymer RD presenting a viscoelastic behavior (Nylon) to let it recover its initial shape. Initial retention should be evaluated as the mean of the 10 first cycles. For fatigue testing, retention force should be reported at specified regular intervals (e.g., 1–10–100–1000–5000–10,000) to reflect the progressive retention loss observed clinically.

Based on this SR, clinical recommendations for AS in IRMO can be made:

For all AS, practitioners can localize their implant according to the clinical situation but should avoid interimplant angulation, especially in the sagittal plane. They need to use dedicated instruments for the maintenance of the RD and favor polymeric RD (CAS) or precious alloy RD (BAS) versus titanium abutments to limit wear of the AS. Recommendations have to be explained to the patient concerning the cleansing and maintenance of prostheses, avoiding aggressive solutions.

CAS should be preferred: (i) in situations of low vertical prosthetic space, although they have a larger cross-section compared to 2.25 mm standard diameter for BAS, (ii) when initial retention is required to be higher, (e.g., in lingual parafunction, bruxism, or in case of a poor surface of sustentation), and (iii) if the axis of the implant is different from the insertion axis of the prosthesis over 30° with the use of extended RD adapted to high angulations. The standard 2.25 mm diameter BAS should be preferred: (i) in situations with low bone width and (ii) if the required retention is not strong but stable over time (e.g., if the patient has difficulty making frequent visits to the practitioner or is disabled). They can be used until 30° angulation in accordance with the insertion pathway.

These recommendations should be seen in the context of other essential aspects such as: (i) the implementation time, and the mastering of the system by the practitioner and the prosthetist, (ii) the easy placement and removal of the prosthesis by the patient, (iii) the inevitable frequent adjustments and repairs, and (iv) the patient compliance for recall.

## Figures and Tables

**Figure 1 materials-15-01933-f001:**
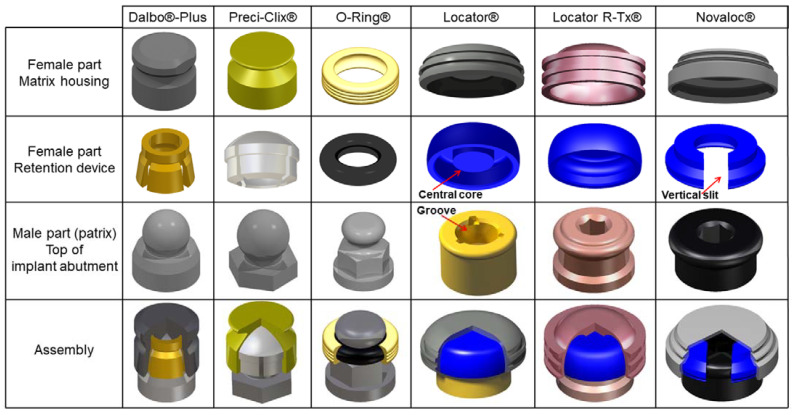
Design of some ball attachment systems (Dalbo^®^-Plus, Preci-Clix^®^, O-Ring^®^) and cylindrical attachment systems (Locator^®^, Locator R-Tx^®^, Novaloc^®^).

**Figure 2 materials-15-01933-f002:**
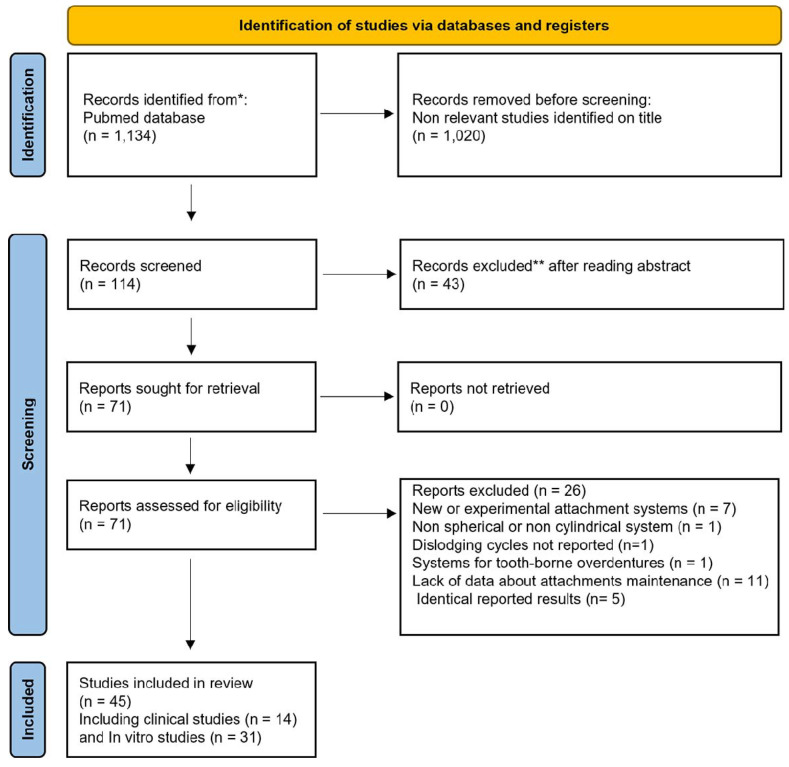
Search method for identification of studies using PRISMA guidelines for systematic reviews.

**Figure 3 materials-15-01933-f003:**
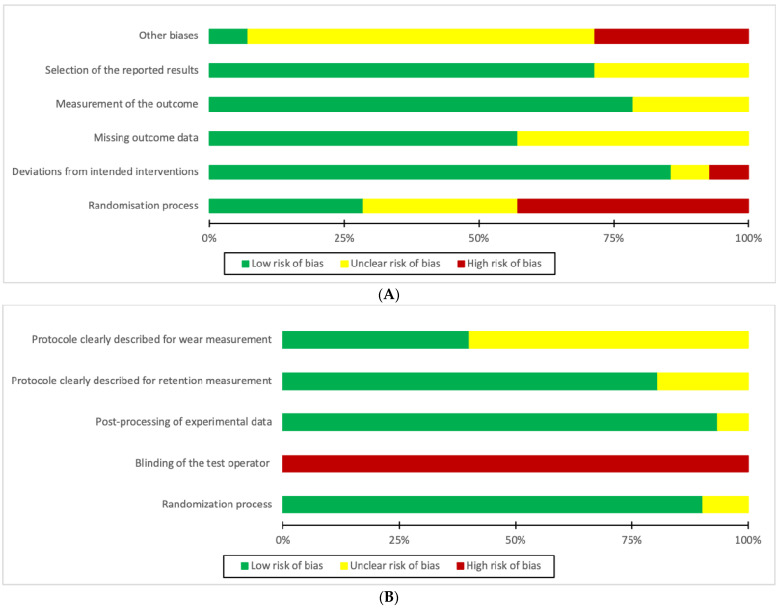
Risk of bias graph: author’s assessment of each risk of bias item in proportions for all clinical (**A**) and in vitro studies (**B**).

**Figure 4 materials-15-01933-f004:**
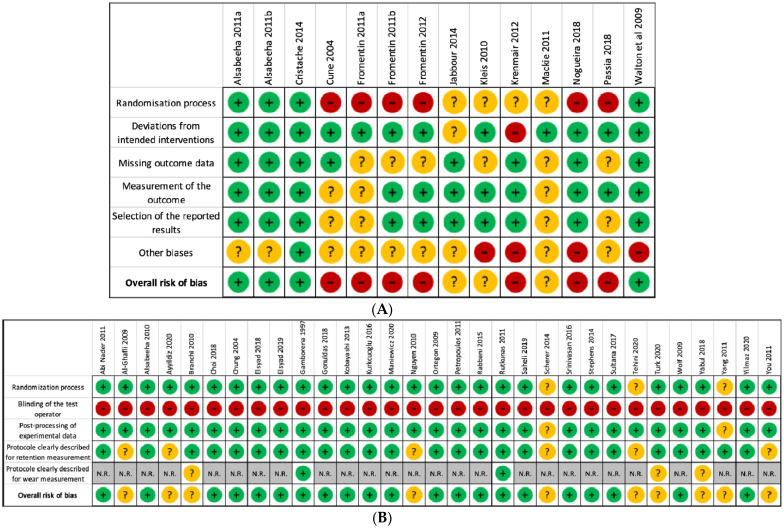
Risk of bias summary: authors’ assessment of each risk of bias item for each included clinical (**A**) and in vitro study (**B**).

**Figure 5 materials-15-01933-f005:**
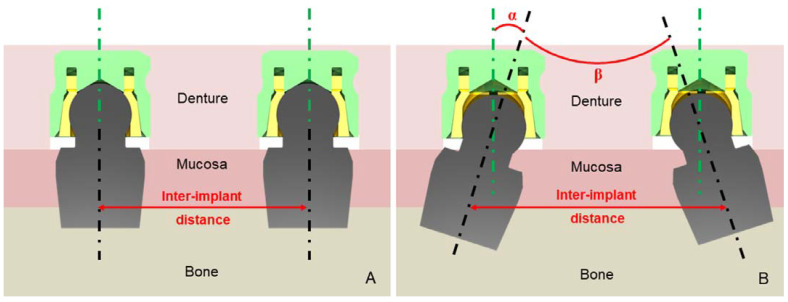
Overdenture with two parallel attachments on (**A**) two parallel implants, and (**B**) two non-parallel implants. Green axis: axis of insertion–removal of the denture. Black axis: axis of the implant. α: implant angulation. β: inter-implant angulation.

**Table 1 materials-15-01933-t001:** Initial retention force (Newton) of ball and cylindrical attachment systems and the influence of insertion–removal (IRC) and/or chewing cycles (CC) on the loss of retention characterized in in vitro studies. Abbreviations: ns, non-significant; Ti, titanium; TiN, titanium nitride. Negative retention loss corresponds to a gain in retention.

Studies	1 or 2-IRMO	Inter-Implant Distance (mm)	Initial and Final Number and Type of Cycles	Cross-Head Speed (mm/min) Medium	Attachments System, Manufacturer	Materials		Retention Setup Before the Test (N)	Retention Force (N)		Retention Loss (%)
						Male Part	Retentive Device		Initial	Final	
Branchi et al., 2010	1	-	1 55000 IRC	54 -	Ball Ø 2.20 mm Sweden & Martina	Ti	Teflon matrix	-	13.92	17.75	27
							Red O-ring^®^ rubber	-	10.20	2.45	75
							Gold	-	15.30	24.42	−50
							Titanium	-	13.24	4.31	68
Yabul et al., 2018	2	19	10 5000 IRC	54 -	Ball, Biohorizons	Ti	Plastic	-	34.12 ± 4.99	11.19 ± 2.8	67.3 ± 5.74
					Ball, DTI, Istantbul		Plastic	-	26.41 ± 5.8	10.58 ± 2.96	58.9 ± 12.21
					Ball, Straumann		Gold alloy	-	48.16 ± 6.46	43.0 ± 6.30	10.5 ± 7.91
					Ball, Straumann		Titanium	-	50.39 ± 4.81	5.59 ± 2.27	88.7 ± 5.11
Elsyad et al., 2016	2	22	1 540 IRC	50 Dry	Locator^®^, Zest Anchors	Ti + TiN	Blue	-	19.64 ± 1.16	14.80 ± 0.83	24.64
							Pink		51.20 ± 0.83	33.60 ± 2.30	34.37
							White		65.20 ± 1.30	39.80 ± 1.48	38.95
Gamborena et al., 1997	1	-	1 5500 IRC	50 Water	ERA^®^, Sterngold	Gold alloy	White		14.91 ± 3.43	2.25 ± 0.49	85
							Orange		24.71 ± 3.14	2.74 ± 0.88	88
							Blue		22.75 ± 2.84	3.73 ± 0.69	85
							Gray		24.03 ± 3.53	3.43 ± 0.98	87
Stephens et al., 2014	2	22	1 5500 IRC	60 Artificial saliva	Locator^®^, Zest Anchors	Ti + TiN	Blue		21.81 ± 7.44	15.97 ± 3.96	5.84 ± 4.45
Choi et al., 2018	2	22	1 400,000 CC 1080 IRC	50 Demineralized water	Locator^®^, Zest Anchors	Ti + TiN	Blue	-	20.1 ± 2.87	20.58 ± 3.09	−2.81 ± 4.07
							Pink		24.55 ± 2.14	37.42 ± 2.79	−52.78 ± 7.78
							White		69.87 ± 5.73	42.56 ± 3.27	38.81 ± 5.32
					CM Loc^®^, Cendres & Métaux	Ti	Light Green		12.45 ± 1.27	12.83 ± 1.39	−3.01 ± 1.92
							Red		17.79 ± 1.70	21.48 ± 1.20	−22.31 ± 17.18
							Green		39.35 ± 3.45	36.99 ± 1.75	5.57 ± 5.28
							Blue		44.07 ± 3.07	43.80 ± 2.45	0.52 ± 1.92
Maniewicz et al., 2020	2	20	1 10,000 IRC	- Artificial saliva	CM Loc^®^, Cendres & Métaux	Ti	Red		81.8 ± 18.5	49.2 ± 12.6	39.85
					Locator R-Tx^®^, Zest Anchors	Ti + TiN	Pink		75.5 ± 24.9	60.0 ± 19.6	20.53
					Novaloc^®^, Straumann AG	Ti + ADLC	Yellow		57.7 ± 31.0	59.4 ± 16.0	−2.95
Yilmaz et al., 2020	2	-	1 1440 IRC	50 -	Locator^®^, Zest Anchors	Ti + TiN	Pink		13.6	10.0	26.5
					Locator R-Tx^®^, Zest Anchors	Ti + TiN	Pink		20.1	14.0	30.4
Tehini et al., 2020	2	22	1 100,000 CC	60 Dry	Locator^®^, Zest Anchors	Ti + TiN	Blue		9.95 ± 1.91	6.37 ± 2.64	−37 ± 0.22
							Pink		15.43 ± 4.08	14.00 ± 3.89	9 ± 0.15
							White		41.73 ± 9.29	38.20 ± 5.11	7 ± 0.12
Chung et al., 2004	2	-	1	50 -	ERA^®^, Sterngold	Gold alloy	White		23.76 ± 1.02		
							Grey		35.24 ± 1.99		
					Locator^®^, Zest Anchors	Ti + TiN	Pink		12.33 ± 1.28		
							White		28.95 ± 0.78		
Jabbour et al., 2014 RCT/crossover	2	-	6 months	Clinical conditions	Retentive anchor, Straumann	Ti	Gold matrix		34.58 *	24.58 *	28.92 *
			12 months							20.27 *	41.38 *
			6 months		Locator^®^, Zest Anchors	Ti + TiN	White	-	39.27 *	15.47 *	60.61 *
			12 months							12.00 *	66.54 *
Gonuldas et al., 2018	2	22	1 2160 IRC	50 Dry	Ball, T.A.G. Medical Products	Ti	Orange plastic cap		30.7 ± 2.82	21.64 ± 1.23	29.51
							White plastic cap		47.9 ± 2.62	26.24 ± 0.71	45.22
					Locator^®^, Zest Anchors	Ti + TiN	Blue		61.39 ± 4.26	5.87 ± 2.42	90.44
							Pink		79.14 ± 3.63	10.4 ± 2.46	86.86
							White		93.75 ± 11.23	18.28 ± 2.25	80.50
Sultana et al., 2017	2	22	10 10,000 IRC	45 Dry	Ball, Dentsply	Ti	Red plastic cap Clix^®^	-	56.2 ± 6.12	46.0 ± 4.74	18.1
					Locator^®^, Zest Anchors	Ti + TiN	Pink		108.9 ± 29.78	20.2 ± 5.74	81.5
							Green		82.3 ± 14.15	17.3 ± 3.73	79.0
Türk et al., 2014	2	22	10 5000 IRC	50 Dry	Ball, O-Ring^®^, Biohorizons	Ti	Plastic cap Clix^®^	-	32.91 ± 5.30	9.70 ± 7.94	69.43 ± 27.61
					Locator^®^, Zest Anchors	Ti + TiN	Pink	-	52.47 ± 6.70	21.70 ± 10.13	57.56 ± 21.65
Wolf et al., 2009	1	-	10 50,000 IRC	- Water	Ball, Dalbo Plus^®^, Cendres & Métaux	Gold alloy	Gold alloy strip	7	8.86 ± 2.2	2.31 ± 1.0	77.54
						Ti	Gold alloy strip	7	9.87 ± 1.6	11.61 ± 3.7	−5.58
					Ball Ecco^®^, Unor	Gold alloy	Green plastic cap	8	6.97 ± 4.6	1.13 ± 0.8	88.55
					Ball Tima^®^, Unor	Gold alloy	Stainless steel ring	8	11.92 ± 3.1	1.55 * ± 1.4	89.82
					Ball, Pro-Snap^®^, Metalor Dental	Gold alloy	Green plastic cap	10	8.52 ± 2.1	3.40 ± 1.5	54.23
					Locator^®^, Zest Anchors	Ti + TiN	Pink	-	13.250 ± 6.6	2.462 ± 1.8	85.66
Alsabeeha et al., 2010	1	-	10 IRC	50 -	Ball Ø 2.25 mm, Southern	Ti	Gold alloy strip	-	17.32 ± 3.68	-	
					Ball Ø 5.9 mm, Southern	Pure Ti + TiN	Plastic cap	-	32.06 ± 2.59		
					Locator^®^, Zest Anchors	Ti + TiN	Blue	-	3.83 ± 0.64		
							Pink		9.40 ± 0.74		
							White		12.39 ± 0.55		
Yang et al., 2011	1	-	1 IRC	60 -	Dal-ro^®^, Biomet 3i	Ti	Gold alloy strip	-	6.48 ± 0.34		
					Locator, Zest Anchors	Ti + TiN	Blue	-	15.36 ± 1.4		
Abi Nader et al., 2011	2	15	1 400,000 CC	15 Dry	Ball Ø 2.25 mm, Nobel Biocare	Ti	Gold alloy	1 turn	10.6 ± 3.6	7.9 ± 4.3	25.47
					Locator^®^, Zest Anchors	Ti + TiN	White	-	66.4 ± 16.0	21.6 ± 17.0	67.47
Petropoulos et al., 2011	2	-	1	50.8 -	Ball Ø 3.5 mm, Nobel Biocare	Ti	Rubber O-ring^®^	-	24.3		
					Ball Ø 2.25 mm, Nobel Biocare	Ti	Ti cap/Ti spring	-	17.8		
					Zest^®^, Zest Anchors	-	-	-	10.8		
					ZAAG^®^, Zest Anchors	-	-	-	37.2		
					ERA^®^, Sterngold	-	White	-	12.7		
							Orange		18.5		
Kobayashi et al., 2014	2	20	10 14,600 IRC	4000 NaCl 0.9%	Retentive Anchor, Straumann	Ti	Gold alloy	7 (0.5 turn)	40.3 ± 15.83	67.9 * ± 15.83	−68.48 (ns)
					Locator^®^, Zest Anchors	Ti + TiN	Blue	-	33.5 ± 9.77	24.57 ± 12.35	26.66
Saheli et al., 2019	2	19	1 1440 IRC	50 -	Ball, Dio Implant, Dio Corp	Ti	-	-	15.6	9.3	40.38
		23							15.3	8.0	46.20
		29							13.8	10.2	26.07
		19			Locator^®^, Zest Anchors	Ti + TiN	-	-	17.1	5.2	69.59
		23							33.5	8.0	76.12
		29							33.1	6.8	79.46
Scherer et al., 2014	2	Inter-canine	1 IRC	50.8	Ball, Zimmer Dental	Ti	White cap	-	35.23	-	
					Ball O-ring^®^ Saturno Standard, Zest Anchors	Ti	Rubber ring		13.13		
					ERA^®^, Sterngold	-	Orange		9.36		
					Locator^®^, Zest Anchors	Ti + TiN	Pink		26.64		
Rutkunas et al., 2011	1		1 15,000 IRC	50 Demineralized water	OP Anchor^®^, Inoue A. Co. Ltd.	Gold alloy	Rubber ring	-	3.15 ± 0.6	3.70 ± 0.2	−17.4
					ERA^®^, Sterngold	Gold alloy	White	-	13.12 ± 3.3	2.89 ± 0.7	87
							Orange		12.63 ± 1.4	2.86 ± 0.5	88
					Locator^®^ Root, Zest Anchors	Stainless steel + TiN	Blue		16.50 ± 9.4	6.24 ± 6.1	62.18
							Pink		15.20 ± 6.9	11.95 ± 3.5	21.38
							White		16.61 ± 2.2	10.28 ± 3.9	38.11

**Table 2 materials-15-01933-t002:** Prosthodontic maintenance of 1- or 2-implant-retained mandibular overdentures and their ball or cylindrical attachment systems. Abbreviations: nc, non-communicated; H, hauteur; RD, retention device; RCT, randomized clinical trial, Ø, diameter.

Clinical Studies Types Follow-Up	Attachment Systems (Male Part/Female Part/Manufacturer)	1 or 2-IRMO	Number of Patients	Maintenance of the Attachment System				Maintenance of the Overdenture			Total Number of Events
				Activated RD	Replaced RD	Replaced Matrix Housing	Tightened or Replaced Patrix	Relined Denture	Fracture or Occlusal Adjustment	Replaced Denture	
Cune et al., 2004 Prospective 1 year	Titanium ball/Gold alloy RD/Friadent, Mannheim, Germany	2	18	7	-	1	1+4	nc	nc	nc	13
Mackie et al., 2011 RCT 8 years	Titanium ball Ø 2.25-mm/Gold RD/Straumann, Basel, Switzerland	2	17	9.6 ± 13.5	1.1 ± 3.2	nc	nc	nc	nc	nc	3.9 ± 2.1
	Titanium ball Ø 2.25-mm/Gold RD/Brånemark, Nobel Biocare		10	18.0 ± 19.8	5.5 ± 7.7	nc	nc	nc	nc	nc	28.8 ± 12.6
	Titanium ball Ø 2.25-mm/Gold-platinum RD/Southern Implants		11	11.9 ± 11.8	2.2 ± 4.4	nc	nc	nc	nc	nc	16.4 ± 7.5
	Titanium ball Ø 2.25-mm/Titanium RD/Straumann, Basel, Switzerland		9	-	13.7 ± 14.7	nc	nc	nc	nc	nc	24.9 ± 10.7
	Titanium ball Ø 3.95-mm/Plastic RD/Southern Implants		22	-	4.3 ± 7.6	nc	nc	nc	nc	nc	8.7 ± 4.2
	Titanium ball Ø 2.2-mm/Rubber ring RD/Steri-Oss (Locator^®^/Zest Anchors after 6 years)		21	-	29.2 ± 24.3	nc	nc	nc	nc	nc	32.2 ± 14.5
Nogueira et al., 2018 Prospective 2 years	Nitrite-coated titanium ball/Nylon RD/Neodent, Curitiba, Paraná, Brazil Titanium ball/Rubber ring/Conexão Sistemas de Prótese, Arujá, Brazil	1	45	-	66	12+7	7+6	3	23	nc	124
Passia et al., 2019 Prospective 10 years	Titanium ball, Camlog Biotechnologies, Switzerland/Gold RD, Dalbo- Plus Elliptic, Cendres & Métaux, Biel, Switzerland	1	11	29	23	nc	5+4	14	8	nc	83
Walton et al., 2009 RCT 1 year	Titanium retentive anchor/Gold RD/Straumann, Basel, Switzerland	1	42	37	4	0	5	60	4+5+2	nc	159
		2	44	34	4	2	1	44	2+2	nc	81
Alsabeeha et al., 2011b RCT 1 year	Uncoated standard titanium ball Ø 2.25-mm/Dalla Bona–type Gold alloy RD/Southern Implants	1	12	13	0	nc	0+2	nc	nc	nc	15
	Titanium nitride–coated ball Ø 5.9-mm/Plastic cap RD/Southern Implants		12	-	2	nc	0	nc	nc	nc	2
	Titanium nitride–coated abutment/Blue nylon cap RD/Locator^®^, Zest Anchors, Escondido, CA, USA		12	-	16	nc	0	nc	nc	nc	16
Cristache et al., 2014 RCT 5 years	Titanium retentive anchor, H 3.4 mm/Gold RD Elitor^®^/Straumann, Basel, Switzerland	2	12	144	8	nc	1	4	2+0	2	161
	Titanium retentive anchor H 3.4 mm/Titanium RD and stainless steel spring, 6.86–10.79 N/Straumann, Basel, Switzerland		11	-	0	0	2	2	1+0	1	6
	Titanium nitride–coated abutment H 3 mm/Pink nylon cap/Locator^®^, Zest Anchors, Inc., Escondido, CA, USA		23	-	22	nc	0	1	0	1	24
Kleis et al., 2010 RCT 1 year	Dal-Ro^®^, Biomet 3i Implant Innovations, Palm Beach Gardens, FL, USA	2	25	4	0	0	1	nc	1	nc	6
	TG-O-Ring^®^ ball/Rubber ring RD/ Cendres & Metaux SA, Biel-Bienne, Switzerland		8	-	10	3	1	nc	0	nc	14
	Locator^®^, Zest Anchors, Escondido, CA, USA		23	-	24	4+4	2	nc	1	nc	35
Krenmair et al., 2012 RCT/crossover 1 year	Ball abutment/Gold RD/Camlog, Screw-line, Altatec)	2	10	2	0	nc	1+1	5	0+1	0	10
	Locator^®^ abutment/Pink RD/Zest Anchors, Inc., Escondido, CA, USA		9	-	4	nc	1	4+1	1	0	11

**Table 3 materials-15-01933-t003:** Wear patterns and dimensional changes of ball and cylindrical attachment systems in clinical and in vitro studies. Abbreviations: RCT, randomized clinical trial; SEM, scanning electron microscope; Ø, diameter.

Studies Types 1- or 2-IRMO Follow-up	Attachment System, Manufacturer	Number of AS	Assessed Parameters/Methods	Results and Location	
				Patrix	Matrix
Alsabeeha et al., 2011b RCT 1-IRMO 1 year	Titanium nitride–coated ball Ø 5.9-mm/Plastic cap RD/Southern Implants	5/12	Wear patterns/SEM Composition of particles/Energy dispersive spectrometer	Unaffected	Slight signs of wear
	Uncoated standard titanium ball Ø 2.25-mm/Dalla Bona–type Gold alloy RD/Southern Implants	5/12		Extensive material loss and abrasion along the path of insertion–removal and across the circumference	Extensive plastic deformation, material flaking and sloughing
	Titanium nitride–coated abutment/Blue nylon cap RD/Locator^®^, Zest Anchors	5/12		Unaffected	Surface rupture and material loss (central core)
Fromentin et al., 2011a Retrospective clinical study 2-IRMO 8 years	Titanium ball anchor/Gold alloy RD Elitor^®^/Straumann	144 (male and female parts)	Wear patterns/SEM Composition of particles/Energy dispersive spectrometer	Year 1: Slight scratches only at the equator Year 3: Slightly deformed profile, scratches at the equator and the summit Year 8: deformed and off-center ball	Year 1: Roughening, and material loss in the form of flakes Year 3: Blunt and deformed lamellae edge along their entire length Year 8: Fatigue cracks or fracture, increased matting of the inner surface, welding of the lamellae
Fromentin et al., 2011b Retrospective clinical study 2-IRMO 8 years	Titanium ball anchor (Ø 2262 to 2267 µm)/Gold alloy RD/Straumann, Basel, Switzerland	69 patrices + 10 controls	Measure of ball Ø, calculation of Ø loss and deviation from circularity in 3 different axes (Vertical V, Mesio-Distal MD, Bucco-Lingual BL)/Coordinate measuring machine with a touch trigger probe	Year 1 (24 AS): Ø 5 to 7 µm (20–23%). Significant more loss in BL but no significant difference in Ø reduction Year 3 (29 AS): Ø 19 to 22 µm (61–91%), significant more loss in BL, 90% at the equator Year 8 (16 AS): Ø 22 to 31 µm. Significant more loss in BL and V	
Fromentin et al., 2012 Retrospective clinical study 2-IRMO 8 years	Titanium ball anchor, Straumann AG, Basel, Switzerland/Gold alloy RD (Ø 2973/2214/2300 µm, E 380/336 µm), Cendres+Metaux, Biel, Switzerland	70 matrices + 10 controls	Measure of the external, internal upper and internal lower matrix Ø and deviations from circularity. Calculation of the upper and lower thickness (E) and the thickness loss of the lamellae tip/Coordinate measuring machine with a touch trigger probe		Year 1 (26 AS): Ø 2989/2232/2298 µm; E 373/342 µm, loss 7 µm. Significant increase of the internal upper Ø, no difference of deviation of circularity in the 3 different areas Year 3 (28 AS): Ø 2937/2282/2309 µm, E 33/316 µm, loss 47 µm. Year 8 (16 AS): Ø 2944/2304/2307 µm, E 310/308 µm, loss 70 µm. In years 3 and 8: increase of the internal upper Ø while the external Ø were significantly lower. Significant increase of deviation from circularity
Jabbour et al., 2014 RCT/Crossover 2-IRMO 1 year x2	Retentive Anchor/Gold RD/Straumann, Burlington, ON, Canada	48	Wear patterns/SEM	No significant scratches, minor flattening (equatorial zone)	nc
	Locator^®^ abutment/White nylon RD/Zest Anchors, Escondido, CA, USA	48	Wear patterns/SEM + high-resolution µCT	No significant scratches and spots	Significant wear and plastic deformation (peripheral notch edge, central core edge)
Abi Nader et al., 2011 In vitro 2-IRMO 400,000 CC (1 year) Dry condition	Titanium ball/Gold lamellae RD/Nobel Biocare	4/16	Wear patterns/SEM	Discrete wear (top and lateral zone part of ball)	Slight wear and discrete deformation, probably due to the activator tool (internal surfaces, corners of retentive lamellae)
	Titanium nitride-coated abutment/nylon RD/Locator^®^, Zest Anchors, Escondido, CA, USA	4/16		Deformation possibly caused by the specific tool (inner surface of the retentive area)	Severe wear (central core and periphery)
Choi et al., 2018 In vitro 2-IRMO 400,000 CC and 1080 IRC (1 year) Deionized water	Pure titanium abutment/Green, Red, Light Green and Blue PEEK RD/CM Loc^®^, Cendres & Métaux	1/20	Wear patterns/SEM	No noticeable abrasion	Slight wear, probably from tools used for placement (top and along the vertical split)
	Titanium nitride–coated abutment/Blue, Pink and White nylon cap RD/Locator^®^, Zest Anchors	1/20		No noticeable abrasion	Severe wear, plastic deformation, surface irregularities, loss of materials more than PEKK RD (retention area, top of central core and periphery)
Gamborena et al., 1997 In vitro 1-IRMO 5500 IRC (3 years) Water	White, Orange, Blue and Grey RD/ERA^®^, APM-Sterngold	3/5	Wear patterns and dimensional Ø changes (top and middle part of central core, metal inner ring of matrix housing)/Traveling three-dimensional microscope		No difference between new and worn matrix housing. Ø loss of central core: −1.80 to +3.54%. The difference (statistical) in Ø between the matrix housing and the middle portion of the RD ranged from 72 to 126 µm
Rabbani et al., 2015 In vitro 2-IRMO 2160 IRC (nc) Artificial saliva	Blue, Pink and White RD/Locator^®^, Zest Anchors, Inc., Escondido, CA	30	Wear patterns/SEM	nc	Significant wear (retention area, central core, periphery)
Rutkunas et al., 2011 In vitro 1-IRMO 15,000 IRC (nc) Demineralized water	Gold alloy ball/rubber ring/OP Anchor^®^, Inoue Attachments Co., Tokyo, Japan	2/5	Wear patterns and dimensional changes/SEM		No Ø loss: 0.52%
	Gold alloy abutment/nylon RD (Orange, White)/ERA^®^, Sterngold, Attleboro, USA	2/5			Ø loss: 0 to 2.48%, smooth surfaces (central core and inner peripherical surface)
	Stainless steel TiN coated abutment/nylon RD (Blue, Pink, White), Locator^®^ Root, Zest Anchors, Escondido, USA	2/5			Ø loss: 0.22 to 5.34%, surface particle loss and irregular surface (central core and inner peripherical)
Stephens et al., 2014 In vitro 5500 IRC (nc) Artificial saliva	Blue RD/Locator^®^, Zest Anchors, Inc., Escondido, CA	20	Wear patterns/SEM	No visible wear	Severe wear (central core > matrix housing)
Türk et al., 2014 In vitro 5000 IRC (nc)	Ball abutment/Rubber ring RD/O-Ring^®^, Biohorizons	10	Dimensional changes (outer and inner diameters of the abutments)/SEM	nc	Ø loss: outer (0.14 ± 0.07mm) and inner (0.08 ± 0.08 mm)
	Locator^®^, Zest Anchors, Inc., Escondido, CA	10		nc	Ø loss: outer (0.17 ± 0.11 mm), inner: 0.11 ± 0.14 mm. No significant difference between the two AS
Wolf et al., 2009 In vitro 1-IRMO 50,000 IRC (nc) Water	Precious alloy ball/Titanium matrix housing with precious alloy RD/Dalbo-Plus^®^, Cendres & Metaux	1	Wear patterns/SEM	Noticeable signs of abrasion	nc
	Titanium ball/Titanium matrix housing with precious alloy RD/Straumann	1		Slight grooves at the equator without any measurable changes in Ø	Minor wear at the tips of the metal lamellae
	Precious alloy ball/Titanium matrix housing with Green plastic RD/Ecco^®^, Unor	1		Large grooves at the equator without any Ø loss	Obvious damages
	Precious alloy ball/Titanium matrix housing with stainless steel spring/Tima^®^, Unor	1		Extensive signs of wear at the equators with Ø loss	Fractures of the retention springs
	Precious alloy ball/Titanium matrix housing with Red plastic RD/Pro-Snap^®^, Metalor	1		Little signs of abrasion at the equator	nc
	Titanium-nickel coating abutment/Stainless steel housing with Pink RD/Locator^®^, Zest Anchor	1		Little abrasion at the equator	Considerable signs of wear
Yabul et al., 2018 In vitro 2-IRMO 5000 IRC (4.5 years)	Ball/Gold RD/Straumann AG, Basel, Switzerland	24	Wear patterns and volumetric loss of ball/Three-dimensional laser scanner	0.7 ± 0.47%	
	Ball/Titanium RD/Straumann AG, Basel, Switzerland	24		25.38 ± 5.41%	
	Ball/Plastic RD/Biohorizons, Birmingham, Alabama	24		12.94 ± 1%	
	Ball/Plastic RD/DTI, Istanbul, Turkey	24		10.47 ± 1.7%	

**Table 4 materials-15-01933-t004:** Influence of implant angulation on the initial retention force and after insertion–removal (IRC) and/or chewing cycles (CC) of selected ball and cylindrical attachment systems. Negative retention loss corresponds to a gain in retention.

In Vitro Studies	1 or 2 IRMO	Inter-implant Distance (mm)	Angulation	Initial and Final Number and Type of Cycles	Cross-Head Speed (mm/min)	Medium	Attachment Systems/Retentive Device	Implant Angulation α (°)	Retention Force (N)		Retention Loss (%)
									Initial	Final	
Ortégon et al., 2009	2	20	Distal	1 3500 IRC	50	nc	Ball Astra-Tech/PreciClix^®^ RD	0 (0 RD)	nc	20.11 ± 2.51	nc
								10 (0 RD)	nc	21.31 ± 1.79	nc
								15 (0 RD)	nc	18.73 ± 4.14	nc
								10 (10 RD)	nc	19.93 ± 1.38	nc
								15 (15 RD)	nc	16.84 ± 1.77	nc
Elsyad et al., 2018	2	22	Mesial	1 540 IRC	50	nc	Locator^®^/Blue	0	20.63 ± 0.70	3.52 ± 0.46	82.94
								5	15.90 ± 0.65	24.02 ± 0.97	51.07
								10	28.07 ± 1.01	26.28 ± 0.62	63.77
								20	47.44 ± 0.51	30.56 ± 0.51	35.58
							Locator^®^/Pink	0	40.07 ± 0.90	19.15 ± 0.78	52.21
								5	30.13 ± 0.82	31.23 ± 0.68	−3.65
								10	31.57 ± 0.51	16.16 ± 1.04	48.81
								20	56.18 ± 0.75	36.45 ± 1.50	35.12
							Locator^®^/White	0	49.20 ± 0.72	32.01 ± 0.01	34.94
								5	40.26 ± 1.10	41.14 ± 1.03	−2.18
								10	41.06 ± 1.00	19.05 ± 0.93	53.60
								20	57.28 ± 0.63	19.99 ± 1.00	65.10
Elsyad et al., 2019	2	22	Distal	1 540 IRC	50	nc	Locator^®^/Blue	0	20.63 ± 0.70	3.52 ± 0.47	82.94
								5	15.29 ± 0.61	9.13 ± 0.81	40.29
								10	30.16 ± 1.25	2.97 ± 1.05	90.15
								20	17.08 ± 0.88	30.26 ± 0.65	−77.17
							Locator^®^/Pink	0	40.07 ± 0.90	19.15 ± 0.78	51.40
								5	34.60 ± 0.53	16.31 ± 1.14	52.86
								10	47.10 ± 0.85	9.11 ± 1.02	80.66
								20	39.12 ± 1.02	21.10 ± 1.01	46.06
							Locator^®^/White	0	48.20 ± 0.72	32.02 ± 1.00	33.57
								5	43.13 ± 1.20	44.01 ± 1.00	−2.04
								10	49.25 ± 1.39	40.18 ± 1.05	18.42
								20	40.30 ± 1.13	41.32 ± 1.50	−2.53
							Locator^®^/Green	20	27.50 ± 0.50	15.42 ± 0.52	43.93
							Locator^®^/Red	20	38.23 ± 1.08	20.14 ± 1.03	47.32
Rabbani et al., 2015	2	23	Mesial	1 2160 IRC	nc	nc	Locator^®^/Blue	0/0	77 ± 13.5	25.8 ± 5.2	65.5 ± 10.2
								0/10	66.4 ± 26.7	14.7 ± 7.9	70.6 ± 22.9
								5/5	73.7 ± 10.1	18.4 ± 3.7	74.1 ± 7.9
							Locator^®^/Pink	0/0	72.7 ± 1.3	27.7 ± 8.2	62.1 ± 10.7
								0/10	74.8 ± 6.7	31.3 ± 4.1	35.1 ± 2.4
								5/5	71.4 ± 3.4	29.4 ± 2.7	30.2 ± 2.7
							Locator^®^/White	0/0	83.8 ± 10.9	32.0 ± 10.7	62.1 ± 9.6
								0/10	101.32 ± 12.0	35.1 ± 2.4	65.1 ± 4.2
								5/5	89.5 ± 15.7	30.2 ± 2.7	65.1 ± 9.4
Stephens et al., 2014	2	22	Distal	1 5500 IRC	60	Artificial saliva	Locator^®^/Blue	0	21.81 ± 7.44	15.97 ± 3.96	26.78
								5	30.03 ± 6.24	15.43 ± 1.59	48.62
								10	24.75 ± 6.83	14.22 ± 2.43	42.54
Al-Ghafi et al., 2009	2	15	nc	1 14,400 IRC	nc	nc	Locator^®^/Green	0	81.75	nc	nc
								5	91.74	nc	nc
								10	104.72	nc	nc
								15	84.86	nc	nc
								20	78.04	nc	nc
Passia et al., 2016	1	-	nc	10 30,000 IRC	nc	Water	Locator^®^/Green	0	21.5	nc	66
								20	24.4	nc	75
							CM Loc^®^/Green	0	22.5	nc	32
								20	27.4	nc	47
Choi et Jeong, 2018	2	22	Distal	1 400,000 CC & 1080 IRC	50	Deionized water	Locator^®^/Blue	0	20.1 ± 2.87	20.58 ± 3.09	−2.81 ± 4.07
								10	22.94 ± 1.48	25.03 ± 2.59	−8.93 ± 5.99
							Locator^®^/Pink	0	24.55 ± 2.14	37.42 ± 2.79	−52.78 ± 7.78
								10	47.13 ± 8.96	30.33 ± 4.18	34.77 ± 6.82
							Locator^®^/White	0	69.87 ± 5.73	42.56 ± 3.27	38.81 ± 5.32
								10	56.86 ± 4.44	39.55 ± 2.95	30.19 ± 5.50
							CM Loc^®^/Green (extralight)	0	12.45 ± 1.27	12.83 ± 1.39	−3.01 ± 1.92
								10	12.11 ± 1.28	15.52 ± 1.41	−30.52 ± 23.84
							CM Loc/Red (light)	0	17.79 ± 1.70	21.48 ± 1.20	−22.31 ± 17.18
								10	27.63 ± 2.28	26.74 ± 2.42	3.25 ± 2.69
							CM Loc^®^/Green (medium)	0	39.35 ± 3.45	36.99 ± 1.75	5.57 ± 5.28
								10	46.96 ± 1.70	43.84 ± 1.99	6.55 ± 5.01
							CM Loc^®^/Blue (high)	0	44.07 ± 3.07	43.80 ± 2.45	0.52 ± 1.92
								10	45.73 ± 4.29	39.44 ± 8.23	14.69 ± 11.22
Manie wicz et al., 2020	2	20	Mesial	1 10,000 IRC	nc	Artificial saliva	CM Loc^®^/Red (light)	0	81.8 ± 18.5	49.2 ± 12.6	39.85
								10	47.1 ± 8.6	42.4 ± 13.5	9.99
								20	47.4 ± 8.9	35.4 ± 9.6	25.32
								30	44.4 ± 10.8	41.2 ± 14.8	7.21
							Novaloc^®^/Yellow	0	57.7 ± 31.0	59.4 ± 16.0	−2.95
								10	41.8 ± 14.8	54.4 ± 15.8	−30.14
								20	48.9 ± 13.9	52.6 ± 8.4	−7.57
								30	76.6 ± 44.5	47.7 ± 18.6	−37.73
							Locator R-Tx^®^/Pink (medium)	0	75.5 ± 24.9	60.0 ± 19.6	20.53
								10	66.3 ± 16.9	45.5 ± 11.9	31.37
								20	62.3 ± 22.1	33.6 ± 11.4	46.07
								30	78.6 ± 33.3	29.9 ± 18.1	61.96
Yilmaz et al., 2020	2	nc	Distal	1 1440 IRC	50	nc	Locator^®^/Pink	0/0	13.6	10.0	26.5
								0/30	18.3	12.8	30.0
								30/30	50.2	21.7	56.8
							Locator R-Tx^®^/Pink	0/0	20.1	14.0	30.3
								0/30	17.5	10.6	39.4
								30/30	33.3	20.0	39.9
Yang et al., 2011	1		nc	1 IRC	60	nc	Dal-ro^®^/Gold alloy strip	0	6.48 ± 0.34		
								15	6.25 ± 0.2		
								30	5.76 ± 0.16		
								45	4.75 ± 0.92		
							Locator^®^/Blue	0	15.36 ± 1.4		
								15	14.67 ± 0.74		
								30	13.3 ± 1.7		
								45	6.58 ± 0.34		
Sultana et al., 2017	2	22	Distal	10 10,000 IRC	50	Dry condition	Ball Dentsply/plastic Red Clix^®^	0	56.2 ± 6.12	46.0 ± 4.74	18.1
								20	45.7 ± 8.03	40.7 ± 2.88	10.9
							Locator^®^/Pink	0	108.9 ± 29.78	20.2 ± 5.74	81.5
							Locator^®^/Green	20	82.3 ± 14.15	17.3 ± 3.73	79.0

**Table 5 materials-15-01933-t005:** Effects of denture cleansing solutions on the initial retention (N) and the retention loss (%) of Locator^®^ attachment system.

In Vitro Studies 1 or 2-IRMO	Cross-Head Speed (mm/min)	Immersion Time, Number and Type of Cycles	RD	Tap Water	Artificial Saliva	NaCl 0.9%	Cool Mint Listerine	NaOCl 6.15%	Aktident	Efferdent	Protefix	Corega	Polident	Polident Overnight
						Sodium Chloride	Menthol Derivatives	Sodium Hypochlorite	Sodium Bicarbonate	Sodium Bicarbonate, Perborate, or Percarbonate, Sodium Carbonate, Potassium Monopersulfate, Citric Acid, EDTA…				
Ayyıldız et al., 2020 2-IRMO	2 50	12 M 1 IRC	Blue	41.1 ± 3.9 N				33.3 ± 4.7 N			44.3 ± 4.1 N	52.5 ± 5.9 N		
			Pink	58.7 ± 6.5 N				39.7 ± 3.8 N			58.5 ± 4.3 N	58.3 ± 6.8 N		
			White	76.7 ± 8.4 N				52.3 ± 8.5 N			89.0 ± 8.7 N	93.7 ± 5.8 N		
Kürkcüoglu et al., 2016 1-IRM0	50	6 M 12 IRC	Blue	22.1 ± 1.2 N				10.4 ± 3.6 N	7.7 ± 2.2 N		13.8 ± 1.5 N			
			Pink	27.3 ± 2.2 N				29.2 ± 3.3 N	25.5 ± 1.5 N		27.6 ± 1.6 N			
			White	36.7 ± 4.0 N				38.3 ± 1.8 N	23.5 ± 2.5 N		33.6 ± 2.7 N			
Nguyen et al., 2010 2-IRMO	50	6 M 1 IRC	Pink	45.3 ± 3.5 N			51.1 ± 5.3 N	7.8 ± 2.5 N		40.8 ± 2.6 N			45.0 ± 2.3 N	45.0 ± 5.2 N
Srinivasan et al., 2016 2-IRMO	120	10 IRC 10,000 IRC	Blue		35.6 ± 7.5 N 29.8 ± 11.1 N	43.3 ± 16.0 N 37.5 ± 11.9 N								
You et al., 2011 1-IRMO	50	6 M 1 IRC 548 IRC	Pink	22.2 ± 2.3 N 10.5 ± 2.9 N 53 ±12% *			22.3 ± 3.1 N 15.8 ± 4.7 N 29 ± 9% *	12.6 ± 1.5 N 7.3 ± 1.0 N 42 ± 11% *		21.5 ± 1.5 N 11.0 ± 2.2 N 49 ± 9% *				21.8 ± 2.4 N 14.4 ± 3.6 N 34 ± 18% *

* Statistically significant retention loss. Abbreviations: CC, chewing cycles; IRC, insertion–removal cycles; M, months; RD, retention device.

## Data Availability

Data sharing not applicable.

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
