# Peer review of "Evaluation of Retention, Wear, and Maintenance of Attachment Systems for Single- or Two-Implant-Retained Mandibular Overdentures: A Systematic Review"

_materials, 2022, doi:10.3390/ma15051933_

Round 1

Reviewer 1 Report

Thank you for your interesting systematic review.
This study performed a comprehensive review between BAS vs CAS.

The study evaluated retention and wear, maintenance of of attachment systems for single or two-implant-retained mandibular overdenture.

Especially, this study reviewed overall Locator-like attachment systems, which would deliver organized information to readers.

The PICO and PRISMA seem to be conducted appropriately. The protocol looks suitable.

With these well-organized protocol, informative tables, and figures, I think this study deserves to be published. 

Author Response

We thank the reviewer for the time spent in revising our manuscript and for his/her positive comments on our work and hope he/she enjoyed reading it. 

Reviewer 2 Report

The research work carried out by the authors is very interesting. The discussion and conclusions are rather flat. The authors should make a large change in the discussion and conclusions to get the manuscript fit for publication. The discussion should be expanded to included a weaknesses paragraph. In particular, in the discussion, in addition to discussing the results obtained, it must analyze them in the light of the experimental evidence and empirical demonstrations brought by other scientists in other scientific articles; this interpretation is carried out on the basis and starting from previous publications on the subject dealt. In addition, references must be written following the instructions for the authors. There are grammatical errors throughout the manuscript.

Author Response

We thank the reviewer for the time spent in revising our manuscript and for the valuable comments provided, which greatly contributed to improve the manuscript.

Considering his/her major comments, a section was added in the manuscript to expose and summarize the limitations/weaknesses of our study (section 3.5 page 10). We have also explained divergent results of some studies by exposing their protocol bias (e.g., page 5, section 3.1, lines 48-49, and page 8, section 3.3, lines 35-39) or other bias (page 7, lines 49-50). To make the conclusion more powerful, clinical and experimental recommendations were extracted from our study and added in the conclusion (page 11, section 4).

Regarding his/her comments about the analysis of the results obtained in our review, we paid attention, for each result, to carefully confront clinical and in vitro studies when possible and to draw a general conclusion. For example, a section analyzing initial retention of BAS vs. CAS is developed page 6, lines 5-17, and we systematically added corresponding values between in vitro mechanical cycles and the estimated clinical wearing time (page 6, section 3.2, lines 35-42, and page 7, section 3.2, lines 17-39). As recommended by the reviewer, we have also checked the references and corrected grammatical errors throughout the manuscript using Grammarly software.

We hope that all our changes, highlighted in yellow in the revised manuscript, answered the reviewer’s comments.

Reviewer 3 Report

  1. The authors mentioned that there are no agreement in literature on a general protocol to perform studies. How this challenge can be tackled to elaborate protocols and improve standards in this field.
  2. I would suggest providing some more details on correlation of in vitro studies to in vivo ones.
  3. Please summarize the criteria to meet to reduce or avoid wear of implant abutments.

Author Response

We thank the reviewer for the time spent in revising our manuscript and for the valuable comments provided, which greatly contributed to improve the manuscript.

Considering his/her major comments, we agree with the reviewer on the fact that there is no agreement in literature on a general protocol to perform in vitro and clinical studies. Regarding clinical studies, it seems impossible to increase standardization because of the variability of clinical situations (patient bone loss, type of prosthetic construction on the opposing arch, dental occlusion, masticatory forces, prosthetic hygiene, and regular dental aftercare visits after setting denture). To obtain comparable data, variability between patients should be drastically reduced. However, if inclusion criteria are too restrictive, it will be impossible to recruit enough patients to obtain the data necessary to draw significant conclusions.

Regarding in vitro studies, lack of standardization is unfortunately very common in mechanical evaluation in dentistry, and we thank the reviewer to give us the opportunity to propose guidelines in our review to improve in vitro testing of AS. These guidelines are exposed in the conclusion (page 11, section 4, lines 11-21).

In this SR, we chose to select both clinical and in vitro of studies to explain clinical observations (e.g., maintenance) from mechanical parameters (e.g., wear or retention) that are primarily measured in vitro. When possible, we directly compared clinical and in vitro findings to draw a general conclusion: for instance, for the wear of BAS (page 7, section 3.2, lines 12-24) or for the comparison of wear and retention loss of BAS vs CAS (page 7 section 3.2 line 47 to page 8 line 8). As suggested by the reviewer, we also added sentences to comment the correlation between in vitro protocols and in vivo studies:

  • Correlation between in vitro mechanical cyclic loading and estimated wearing time (page 6, section 3.2, lines 35-42).
  • Comments about in vitro control of insertion/disinsertion axis and implant angulation (page 8, section 3.3, lines 35-39)
  • Comments on cyclic in vitro loading compared to in vivo loading (page 9, section 3.4, lines 40-42).

 We also agree with the reviewer that recommendations to reduce the wear of the AS were given at several places in the manuscript (page 6, section 3.2 lines 45-47, page 8 section 3.2, lines 9-11, page 9, section 3.4, lines 31-36…). A clear summary of the criteria to meet to reduce the wear of the AS has been added in the clinical recommendations, in the conclusion (page 11, section 4, lines 24-28).

We hope that all our changes, highlighted in yellow in the revised manuscript, answered the reviewer’s comments.

Reviewer 4 Report

Review Materials Manuscript 1562791

Evaluation of Retention, Wear and Maintenance of Attachment Systems for Single or Two-Implant-Retained Mandibular Overdentures: a Systematic Review

The article is interesting and has potential for publication at Materials, but, some major points should be considered before acceptance. These are detailed as follows:

Introduction

This section is well-written and informative, and the explanations about the attachment systems were adequate and valid to improve the article's comprehension. Figure 1 is valid and informative, as well.

Materials and Methods

This section, overall, is well written. However, in my humble opinion, since this is a Systematic Review, some considerations must be made concerning the design of the review. Usually, a Systematic Review is more effective in finding valid conclusions when the outcomes and variables are more specific. In the present review, both in vitro and in vivo studies were considered valid, although they were compared separately during the risk of bias analysis, which is adequate. The authors could explain why they did not perform 2 separate reviews (one comparing the in vivo studies and one the in vitro).

The PICO question comprises 5 different outcomes (Initial retention, Retention after clinical use, Retention loss, Wear, Maintenance), and it is really difficult to find all these aspects in a single study. When the tables are checked is easy to understand why there is so missing information (the variability of studies designs and the heterogeneity of data is enormous). In my humble opinion, this is a limitation of the review and should be better discussed at the end of the Results and Discussion section.

It is clear that evidence from a systematic review or meta-analysis of relevant RCTs (randomized controlled trial) is considered the top of the Evidence-Based Practice pyramid. However, systematic reviews, no matter how well conducted, cannot overcome the methodological flaws and limitations in their included primary studies. Considering this, other strategies such as the use of the Grading of Recommendations Assessment, Development, and Evaluation (GRADE) approach to assess the certainty of evidence from clinical trials is valid. Parameters such as the risk of bias, inconsistency, indirectness, imprecision, and publication bias should be considered to assess the quality of evidence. Authors are encouraged to perform it or justify the absence of this as a limitation of the review, in the Results and Discussion section.

Results and Discussion

The section is complete, well written, and informative, and adequate literature has been used to perform the comparisons. In addition to the suggestions previously made, authors are encouraged to suggest, at the end of the Results and Discussion section, how variables and designs future primary studies should evaluate, to improve the determination of the best Evidence-Based Practice information on the theme.

After this major review, the article should be checked again to evaluate the possibility of publication.

Author Response

We thank the reviewer for the time spent in revising our manuscript and for the valuable comments provided, which greatly contributed to improve the manuscript.

Regarding his/her comment about inclusion of both clinical and in vitro studies, we made that choice to explain clinical observations (e.g., maintenance) from mechanical parameters (e.g., wear or retention) that are primarily measured in vitro. When possible, we directly compared clinical and in vitro findings to draw a general conclusion: for instance, for the wear of BAS (page 7, section 3.2, lines 12-24) or for the comparison of wear and retention loss of BAS vs CAS (page 7, section 3.2, line 47 to page 8, line 8). In our opinion, this systematic comparison improved the quality of evidence of the review by providing two different points of view on the same subject and enabled the authors to better highlight the weaknesses of studies presenting contradictory findings. As suggested, a section was added to expose the limitations of our study (page 10, section 3.5), and in particular lines 11-15 and lines 34-36 provide an answer to this comment.

We agree with the reviewer that data heterogeneity is a limitation of our study. Corresponding comments were added in page 10, section 3.5, lines 15-23 and 31-33.

GRADE assessment was not possible secondary to the inability to conduct a meta-analysis. This limitation was added in a new section (page 10, section 3.5, lines 27-29).

To improve the determination of the best Evidence-Based Practice information on the theme of our SR, we suggest a standardization of their protocols.

Regarding in vitro studies, lack of standardization is unfortunately very common in mechanical evaluation in dentistry. We thank the reviewer for his/her comment, giving us the opportunity to propose guidelines to improve in vitro testing of AS in the conclusion (page 11, section 4, lines 11-21).

Regarding clinical studies, it seems impossible to increase standardization because of the variability of clinical situations (patient bone loss, type of prosthetic construction on the opposing arch, dental occlusion, masticatory forces, prosthetic hygiene, and regular dental aftercare visits after setting denture). To obtain comparable data, variability between patients should be drastically reduced. However, if inclusion criteria are too restrictive, it will be impossible to recruit enough patients to obtain the data necessary to draw significant conclusions. To help clinicians improve their practice, we finally decided to add clinical recommendations in the conclusion (page 11, section 4, lines 23-38) based on the primary findings of this SR, supported by clinical and in vitro studies.

We hope that all our changes, highlighted in yellow in the revised manuscript, answered the reviewer’s comments.

Reviewer 5 Report

General comment: This paper is well written and organized! In general, the authors have described the appropriate terms correctly, but in some places, they are not clear. English needs to be revised, because grammatical errors are present while reading the work. Please find critical comments as follows:

Critical comments:

Running title: The referee suggests removing it, because the main title details the following manuscript. In this manner creates confusion when the reader starts from the title to focus. Why the authors need to put the short title below?

Summary box: Furthermore, this section is not necessary to be maintained. Because the conclusion contains this and in a sense the authors repeat the things. Here, the second sentence is not clear in the way it is described.

Abstract: ROB2, this word needs its long name first and then putting its acronym into brackets (ROB2); “In vitro” must be in italics, not only here but throughout the paper, because there are places where this word is not expressed in italics; The last sentence is not written correctly in English.

Point 1.1.: Here, please put the manufacturer, city and state of Dalbo® Plus. Not only for this, but also for the other materials mentioned in the paper and analyzed.

Point 3.4.: Sodium bicarbonate, here it must be in lowercase.

Author Response

We thank the reviewer for the time spent in revising our manuscript and for the valuable comments provided, which greatly contributed to improving the manuscript.

We have therefore removed the running title and the summary box. We added the long name for RoB2 and used Grammarly software to correct grammatical errors throughout the text. Concerning the other critical comments, we have made all the requested changes: page 2, section 1.1 and page 9, section 3.4, line 31.

We hope that all our changes, highlighted in yellow in the revised manuscript, answered the reviewer’s comments.

Round 2

Reviewer 2 Report

All issues have been addressed. No further suggestions or comments.

Reviewer 4 Report

The authors have considered the suggestions made and answered all the questions, which improved the article in a significant way. The considerations about the study limitations and suggestions for future studies are adequate and important.

I consider that the study is adequate for publication in the present form